# Advanced Tumor Imaging Approaches in Human Tumors

**DOI:** 10.3390/cancers14061549

**Published:** 2022-03-18

**Authors:** Samuel Nussbaum, Mira Shoukry, Mohammed Ali Ashary, Ali Abbaszadeh Kasbi, Mizba Baksh, Emmanuel Gabriel

**Affiliations:** Department of Surgery, Section of Surgical Oncology, Mayo Clinic, Jacksonville, FL 32224, USA; shoukry.mira@mayo.edu (M.S.); maashary97@hotmail.com (M.A.A.); abbaszadehkasbi.ali@mayo.edu (A.A.K.); baksh.mizba@gmail.com (M.B.); gabriel.emmanuel@mayo.edu (E.G.)

**Keywords:** tumor imaging, artificial intelligence, molecular imaging, intravital microscopy

## Abstract

**Simple Summary:**

This review highlights several recent advances in cancer imaging, including artificial intelligence, molecular imaging, and intravital imaging. We discuss these areas’ potential to change the detection, monitoring, and overall treatment of cancer.

**Abstract:**

The management of cancer has always relied heavily on the imaging modalities used to detect and monitor it. While many of these modalities have been around for decades, the technology surrounding them is always improving, and much has been discovered in recent years about the nature of tumors because of this. There have been several areas that have aided those discoveries. The use of artificial intelligence has already helped immensely in the quality of images taken but has not yet been widely implemented in clinical settings. Molecular imaging has proven to be useful in diagnosing different types of cancers based on the specificity of the probes/contrast agents used. Intravital imaging has already uncovered new information regarding the heterogeneity of the tumor vasculature. These three areas have provided a lot of useful information for the diagnosis and treatment of cancer, but further research and development in human trials is necessary to allow these techniques to fully utilize the information obtained thus far.

## 1. Introduction

Technological applications in medicine are constantly evolving and progressing to improve outcomes for patients. This is especially prevalent in the disciplines of medical and surgical oncology where novel imaging modalities are increasingly being developed. Understanding the optimal utilization of these innovative techniques in conjunction with the established conventional methods is of the utmost importance in the treatment of cancer. In this review, we highlight three different areas where tumor imaging approaches have had significant advancement in recent years and how continuing to improve these areas may impact the detection and treatment of cancers in the future. Limitations and obstacles to further development and application are also discussed. The areas of technological development and tumor imaging methods that have made great strides in cancer treatment are artificial intelligence, molecular imaging, and real-time intravital imaging.

## 2. Artificial Intelligence

With current imaging modalities such as computed tomography (CT) and magnetic resonance imaging (MRI) constantly improving in quality, it is essential to also improve upon the methods of imaging interpretation such that patients may benefit from these advances. Throughout recent years, artificial intelligence (AI) has emerged as a potential tool to interpret imaging and thus aid in the detection, characterization, and monitoring of cancers.

While conventional imaging interpretation typically involves the analysis of qualitative features such as tumor density, size, margins, and anatomical relationships by a trained radiologist, advanced AI systems instead work by quantifying tumor characteristics [1]. Deep learning, a subset of machine learning, has recently become a major contributor towards the increasing use of AI in clinical settings. This rapidly advancing branch of machine learning uses layers of artificial neural networks (ANN) intended to better mimic human performance. Despite being rather low maintenance in the long run, deep learning systems are created in a process that involves presenting the computer with large volumes of unstructured data [2]. Contrary to older approaches to machine learning which typically require structured data along with ongoing human supervision to encourage correct pattern recognition, these multi-layered ANNs require minimal ongoing intervention. Another advantage of deep learning systems, as opposed to conventional machine learning, lies in the ability to instantly yield reliable results which constantly improve in quality as they are presented with more data. For these reasons, deep learning has become a promising area of research in medicine. 

Working alongside expert radiologists, AI has the potential to be a valuable clinical tool in the setting of cancer imaging. The implementation of AI should not be seen as an alternative to expert radiologists, but rather as a collaboration between the two. In fact, one of the earliest documentations of AI use for cancer detection, published in 1991, suggested that a hybrid approach incorporating both neural networks and clinicians appeared to be the most promising [3]. Additionally, while recent advances suggest that it is possible to close the gap in performance between radiologists and machines, AI performance is yet to exceed that of trained professionals. A study by Rodriguez-Ruiz et al. tested this concept in a study comparing the performance of deep learning AI systems with that of 101 radiologists. Both groups independently analyzed 2652 cases of digital mammograms and produced a score representing suspicion of malignancy for each case. The study found that, while it showed comparable results to the average of all included radiologists, AI performance remained lower than the best radiologist [4]. Although ongoing large-scale studies continue to assess the reliability of AI as a stand-alone tool for imaging interpretation, the conversation has focused on how AI can supplement the work of physicians. Studies have found that it is an incredibly useful tool when used as a “second opinion”. For instance, studies have assessed whether trained radiologists can better assess the likelihood of malignancy on standard mammographs with the help of computer-aided detection. Interestingly, not only did the average sensitivity increase significantly, but it was found that radiologists were more likely to order biopsies for malignant masses with the help of the computer estimate [5].

Strides towards AI use in clinical settings have not been limited to mammography but continue to expand to several imaging modalities. Recent studies have noted an increase in lesion detection on breast ultrasound with AI assistance when compared to radiologists alone [6]. Similar results were observed with skin cancer screening in the analysis of dermoscopic imaging [7]. Additionally, AI assistance has been observed to improve the contouring accuracy for nasopharyngeal carcinoma on MRI [8]. AI may even be used as a tool to assist with the interpretation of biopsies. For instance, deep learning has been employed for the detection and grading of prostate cancer on whole slide images of core needle biopsies [9]. More recently, AI has also been employed to aid in the interpretation of positron emission tomography scans in the setting of various cancers such as prostate or head and neck [10,11]. These studies demonstrate that researchers have only scratched the surface of what AI is capable of in terms of imaging interpretation. 

While recent years have seen an increase in the use of AI in oncology practices, it is yet to be widely implemented, which represents a significant challenge to the acceptance of AI in clinical practice. This has partially been credited to the lack of available high-quality data that can be used to train deep learning systems [12]. Additionally, developers must ensure that the dataset presented to the AI systems is heterogeneous to prevent unintentional bias, meaning that the data must be representative of the general population and include sufficient examples of underrepresented populations [10]. Any bias in this aspect may lead to unreliable treatment recommendations for such populations. Another potential issue preventing the widespread adoption of AI is that it needs to be seamlessly integrated into the physicians’ workspace in a way that does not disrupt it. According to a survey of MRI technologists, incorporating information technology and AI training into the radiology curriculum remains one of the biggest challenges. While 91.8% of surveyed technologists understood the potential benefits of AI in their practice, only 29.6% have been able to be educated on the topic [13]. However, with increased availability of data and training, AI may become an exciting addition that helps reduce scan times and radiologist workload in the near future.

## 3. Molecular Imaging

Molecular imaging is crucial to the detection and management of cancer and advances in this technology may increase the precision with which therapies can be implemented [14]. Because the tumor microenvironment represents a complex physical and biochemical system that is involved in tumor initiation, progression, metastasis, and drug resistance, being able to utilize molecular imaging to visualize biomarkers at these checkpoints as well as the cancer therapies is critical to the ability of oncologists to make effective treatment decisions [15]. Therefore, it is important to understand how molecular imaging is effective and what enhancing these techniques may accomplish in terms of detecting and managing cancers. 

The effectiveness of molecular imaging relies heavily on the imaging modalities and probes/contrast agents used to target, detect, and visualize cancer biomarkers. Imaging modalities include magnetic resonance imaging (MRI), positron emission tomography (PET), or single photon emission computed tomography but may also use computed tomography (CT), and ultrasound sonography (US). Imaging agents are primarily comprised of target-specific molecules (e.g., small molecules, peptides, antibodies) to recognize and bind to TME biomarkers, reporters (paramagnetic substances, radionuclides, and fluorophores) that are visible in different imaging modalities, and spacers/carriers that connect the ligand and reporter together [15]. There is some debate as to whether MRI or PET is the most accurate modality on its own, though each have limitations alone. As such, combination/hybrid imaging techniques like PET/CT or PET/MRI have been examined to see if they are superior to conventional imaging (MRI, CT) in detecting cancers. There is evidence that this may be true for some tumors, as one study looked to see if 18F-fluorodeoxyglucose (18F-FDG) PET in addition to MRI is more effective at diagnosing primary tumor staging of cervical cancers [16]. The results of this study found that combining the diagnostic advantages of these modalities enables improved treatment planning. However, the metabolic agents used may be cancer and biomarker specific as seen by 18F-FDG’s sensitivity in part to the glycolytic activity of malignant cells, and therefore may not be appropriate in prostate cancers for instance. Still, it was found that MRI combined with PET using different receptor-targeting agents was more effective in diagnosing prostate cancers than MRI alone [17]. 

Using different combinations of contrast agents and imaging modalities may be appropriate for diagnosing and treating different cancers. Because tumors are often characterized by genomically and phenotypically distinct cancer cell subpopulations (known as tumor heterogeneity), greater imaging contrast and higher specificity for cancerous tissues are sometimes required, which has led to the use of two targeting ligands [18,19]. While this dual targeting strategy may enable clearer visualization of cancer lesions, especially in heterogenous tumors, it still is not appropriate in all situations because dual targeting may make it more challenging to differentiate which antigen is being recognized. This in turn may pose limitations in the interpretation of complex molecular imaging techniques [18]. Nonetheless, molecular imaging is a very useful tool in detecting and treating cancer but knowing which techniques to use and how to best utilize them is critical.

Much progress has been made in recent years to develop and improve molecular imaging for the detection and therapy of cancers, and still many of these new techniques are being researched currently. One area of molecular imaging that has recently been investigated is the tracking of immunotherapy and the interaction between the tumor microenvironment and immune system. The role of the immune system in cancer therapy has long been appreciated. Current research is looking into using molecular imaging to provide unique insight into the mechanisms of cancer immunotherapy and track the immunological responses to evaluate and, in the future, enable targeted therapy without inducing adverse events [20]. Real-time, clinical characterization of the tumor microenvironment may provide important information in a personalized approach to determine which patients may benefit from different immunotherapies, such as immune checkpoint blockade or adoptively transferred effector cells.

Another area in which there has been much interest is the development of imaging probes and contrast agents to identify molecular targets linked with the manifestation of disease. One such technique is the use of Raman Spectroscopy, which provides highly specific information pertaining to the molecular bonds in amino acids, lipids, proteins, and nucleic acids, by detecting inelastically scattered light following tissue excitation with a near-infrared laser [21]. Early results have shown the ability to detect margins between different types of tissue based on the protein to lipid ratio, including between white/gray matter, muscle/adipose tissue, as well as between cancerous and non-cancerous tissue. However, this technique still needs to be refined as there can be differences in the spectra obtained that contribute to less clearly defined images when imaging deeper tissues. Additionally, current Raman probes are limited to single point analysis, whereas increasing this capability may allow for a more complete visualization of the tumor extent, particularly as it relates to the margins [21]. The use of gold nanoparticles (AuNP) as a probe has also shown promise in this manner due to their ease of preparation, stability in aqueous media, biocompatibility, and strength of contrast enhancement in various imaging modalities. These AuNPs have shown encouraging results in the use of molecular imaging both on a cellular level as a tracker and as a contrast agent. However, there are only a few AuNPs being investigated clinically because of questions surrounding their systemic toxicity, long-term accumulation, and excretion from the body. It is anticipated that the research into AuNPs will focus on the excretion of these particles to bring the benefits of AuNPs to clinical imaging [22]. 

Another application of molecular imaging on the cutting edge of research is in molecular imaging guided surgery. Only recently has there been information regarding the direct use of molecular imaging in the operating room, whereas surgeons have traditionally relied on visual and tactile tissue assessment to make surgical decisions. Improved adaptations of existing imaging technologies have allowed molecular imaging guided surgery to become more prevalent through the use of nuclear and optical imaging. Nuclear imaging allows quantitative deep-tissue imaging and is adaptable for the rapid assessment of more superficial tissue, while optical imaging enables high-resolution real time imaging at shallower depths. As these techniques mature, combining their strengths in a multimodal approach may be able to deliver improved surgical outcomes and may allow molecular imaging guided surgery to play a vital role as a state-of-the-art surgical practice [14]. The amount of progress that molecular imaging has gained in recent years has been significant, and the additional improvement on these existing technologies is expected to make cancer treatments more effective.

## 4. Real-Time Intravital Imaging Methods

Intravital microscopy (IVM) is an imaging technique that allows microscopic observation of living tissue and organic processes in vivo in real time and is an important tool to utilize when visualizing the tumor microenvironment and its associated vasculature. Compared to conventional microscopic examination of tissue and other in vitro methodologies like histopathology, immunohistochemistry, or flow cytometry, IVM allows for a more comprehensive understanding of biologic processes at more than just static snapshots in time and is therefore a powerful tool to be used to characterize and treat tumors [23]. As such, it is important to know what information has been gained so far from this technology and how further development may impact the management of cancers.

IVM’s utility mainly lies in its ability to deliver real-time direct observations of the tumor-associated vasculature as well as the tumor microenvironment. While IVM is a relatively old technique, it has stayed current by evolving along with conventional microscopy and has been combined with other visualization technology like bright field, single photon fluorescent imaging, confocal microscopy, and multiphoton microscopy [24]. This has allowed the technique to have a higher resolution at the cellular and sub-cellular layer as opposed to the organ layer like other imaging modalities such as CT or MRI, which makes it useful in tracking tumor metabolism, remodeling, and angiogenesis [23]. These techniques can be used in different scenarios to best utilize their unique benefits. For example, similar to intraoperative molecular imaging, intraoperative confocal microscopy and confocal endomicroscopy have been used to look at tumor margins and tissue dysplasia respectively in several types of cancers. Multiphoton imaging, which has deeper penetration and decreased light scattering compared to confocal microscopy, has been used to observe cancer cell motility [24,25]. One of the most important things that IVM has discovered is that the tumor vessels lack the stratification and hierarchy like that of normal vessels in that arterioles, capillaries, and venules cannot be distinguished within tumor tissues, and, as such, these vessels have irregular diameters, aberrant branching patterns, abnormal blood flow rates, and anastomotic strictures [26]. This abnormal vasculature plays a major role in the development of the factors that make the tumor microenvironment hostile like hypoxia, elevated interstitial fluid pressure, and extracellular acidoses and has been shown to cause resistance to several treatment modalities while inducing malignant progression, invasive growth, and metastatic spread [27]. In addition to the vasculature, IVM has also provided information regarding the relationship between the tumor microenvironment and the immune system. Using these techniques, significant insights have been made into the immune mechanisms of leukocyte migration/trafficking into lymph nodes and tumors as well as lymphocyte activation or effector immune responses [23,28]. However, many of the discoveries that have been made thus far using IVM has come in investigating preclinical tumor animal models as opposed to in human cancers, though new human studies are becoming more frequent. Even so, the data that have been gathered so far from IVM have significantly increased the ability to characterize and understand cancers.

While the use of IVM has elucidated many tumor characteristics that were previously unknown, typically from preclinical animal trials, ongoing and future studies will hopefully provide better treatment information and include looking into more human cancers. The abnormal vasculature of tumors has been confirmed to be present from a recent study of a human tumor peritoneal carcinomatosis, and further study of these implications could greatly influence the effectiveness of systemic drug delivery of chemo/immunotherapies to the tumor microenvironment [29]. The associations between IVM tumor vessel measurements and tumor response to neoadjuvant therapy were investigated, and while the results were not significant, the low sample size will be addressed in future studies, which should further the ability to use IVM tumor observations in cancer treatment. There is another ongoing study trying to determine the feasibility of IVM in patients with solid tumors during standard surgical resection including when vasopressors or fluid boluses are administered, and while the results have not yet been quantified, it is expected that the trial will support the development of IVM technologies to improve patient treatment [26]. This study hopes to provide a foundation on which IVM can be used to predict and even augment the clinical response to systemically delivered therapies based off of the detected blood flow parameters. 

There is also anticipation that this technology can be used to track immune responses, and more specifically, T-cell activation. While much has already been learned about antigenic activation and T-cell biology from IVM, including the motility of T-cells, the existence of distinct phases of interaction with dendritic cells during T-cell priming, and much more, there are still a lot of unknowns regarding this process including the correlation between early activating signals and amplitude/response in a single T-cell. As such, further in vivo studies may give great insights into the way in which the body is protected from infections and tumors by the adaptive immune system [30]. The use of IVM has shed light on numerous tumor characteristics already and should lead to a better understanding of how to best utilize therapies to treat them in the future.

## 5. Discussion

Developing innovative methods to improve the detection, monitoring, and treatment of cancer has been at the forefront of medical research, and, as discussed in this review, much has been learned about the nature of human tumors in recent years due to the advancements in artificial intelligence, molecular imaging, and intravital imaging. While many of the imaging modalities used to observe tumors like MRI, CT, and PET have been present for decades, the quality of these instruments has always been improving since their inception and continue to do so. 

The use of artificial intelligence, molecular imaging, and intravital imaging represent approaches that have made significant progress in recent years, and perhaps the most innovative approaches are integrating these tools together to optimize diagnosis and subsequent treatment of cancer. Artificial intelligence has exhibited an ability to quantify tumor characteristics by analyzing large volumes of unstructured data using artificial neural networks that become increasingly accurate as more data are acquired. Though AI has yet to supplant the performance of physicians in general, integration of AI in a manner that does not disrupt existing practices may provide the best results. Recent advances in AI have also influenced the progression of molecular imaging and intravital microscopy, from image acquisition and quality to diagnosis, and as such have aided many of the discoveries that came from them [30,31]. However, many of the molecular imaging applications of AI have been isolated to investigational research and are only now branching into the clinical space through commercialization and integration into scanner hardware [31]. Despite this, AI and molecular imaging are being developed into useful techniques in diagnosing different types of cancers by utilizing various modalities and probes/contrast agents depending on the type of cancer being investigated. In addition, AI has crossed into the field of intravital microscopy with applications that integrate with the classification and segmentation of microscopy images, label prediction from label-free images, resolution enhancement, de-noising of images, and recovery of isotropic resolution [32]. This has helped IVM obtain significant insights into the stratification of the tumor microenvironment that were previously unknown. Table 1 summarizes the major findings of one source from each of the three areas. These improvements in the technology and techniques used to observe tumors have made huge strides in how physicians diagnose and treat cancer.

Despite having learned a great deal about the nature and physiology of cancer in recent years due to improved techniques, much of the development and utilization of these approaches in the future hinges on the continual education and research into these topics with an emphasis on developing more human trials. Many of the advances made in the areas of AI, molecular imaging, and IVM have come from studies into animal models and have yet to be implemented and investigated in humans. There are still many challenges that these techniques will need to overcome to fully realize their potential. AI is just now maturing and is not quite reliable at the level that physicians are comfortable in incorporating it into their practice. This will most likely require an abundance of proof of efficacy from prospective studies. Within this, the goal of current research into AI has focused on improving performance of these models, but going forward, this may shift to showing transparency and accountability in order to identify shortcomings and successes [1]. 

Molecular imaging is also affected by the progress made by AI as the former has the potential to be used to discover imaging biomarkers associated with tumor responses to treatments. In addition, because molecular imaging relies heavily on the imaging probes/contrast agents that are used, research into new agents takes time because they require Investigational New Drug applications to the FDA to prove safety [33]. The challenges that face IVM, specifically two-photon microscopy, may also be alleviated with improvements in artificial intelligence. One of the biggest things that IVM will need to overcome is the tissue motion caused by natural rhythms within the body such as heartbeat, respiratory cycles, and peristalsis. This motion can impede temporal and spatial resolution and is variable for different organs, different patients (due to body habitus for example), and different types of anesthesia. This issue is also not one that can be accurately studied in vitro, making human trials all the more necessary. Increases in computational resources and calculation power that may come with more advanced AI should help the active motion compensation to make image acquisition better [34].

## 6. Conclusions

In each of the innovative techniques that we have reviewed, further ongoing research is required to more successfully bring these tools to the clinical practice and substantiate everyday clinical utility. Development of these techniques individually and investigation into their integrated benefits will likely accomplish these goals.

## Figures and Tables

**Table 1 cancers-14-01549-t001:** Significant Sources.

Date	Study Name	Major Findings
September 2019	Stand-Alone Artificial Intelligence for Breast Cancer Detection in Mammography: Comparison With 101 Radiologists	Plotted on an AUC-ROC curve, the AI system alone was statistically noninferior to the 101 radiologists who analyzed the same datasets of digital mammography (DM) AUC difference was 0.026 The AI system had a higher AUC than 61.4% of radiologists and a higher sensitivity than 57.9% of radiologists Though the AI seemed to outperform the radiologists, the average performance score of radiologists was very close and AI was still not comparable or superior to the best radiologist Utilizing AI seems to be feasible in the scenario studied though there are many scenarios that may have skewed this finding including the AI having no prior knowledge of previous images, tumor type, etc.
August 2017	Comparison of ^18^FDG-PET/MRI and MRI for pre-therapeutic tumor staging of patients with primary cancer of uterine cervix	There was no statistical difference for the detection of tumor invasion of adjacent organs/tissues within the female pelvis MRI correctly determined the T stage of the patient cohort (all with varying stages) 87% of the time as opposed to 85% for PET/MRI Both were found to have underestimated the same number of cases (2 out of 53 cases) and Both modalities overestimated stage for one patient due to 18-FDG accumulation skewing the interpretation PET/MRI correctly identified lymph node involvement in a higher number of patients than MRI alone PET/MRI was also superior in detecting metastatic spread to pelvic or paraaortic lymph nodes PET/MRI was better able to identify metastatic spread than MRI (87% and 67% respectively) Therapeutic decisions of the simulated interdisciplinary tumor board were influenced by PET/MRI due to its false identification of tumor stage previously stated, and of note, MRI alone could not correctly diagnose these particular 2 cases PET/MRI correctly identified simultaneous breast cancer in one patient which was not found by MRI MRI seems to be an adequate modality though data shows that 18F-FDG PET can provide valuable additional information to help guide treatment such as tumor metabolism Combining the modalities proves to valuable in the primary tumor staging of cervical cancer patients
March 2021	A pilot trial of intravital microscopy in the study of the tumor vasculature of patients with peritoneal carcinomatosis	Human intravital microscopy (HIVM) demonstrated statistical differences between the tumor and control fields among vessel measurements except for mean non-functional vessel diameters Tumor-associated areas were shown to have lower density of functional vessels, higher density of non-functional vessels, and higher proportion of non-functional vessels compared to non-tumor controls Tumor vessels had a significantly smaller mean diameter in tumor areas as opposed to non-tumor areas Non-functional vessel diameter was similar between tumor and non-tumor areas Mean blood flow velocity of functional vessels within tumor areas was significantly slower than mean velocity of functional vessels within non-tumor areas When treated with neoadjuvant therapy, similar results were shown to those stated above Real-time HIVM images demonstrated high proportion of normal, streamlined blood vessels in non-tumor associated vessels when compared There were not statistical associations between the HIVM vessel characteristics and patients’ response to neoadjuvant therapy Despite no association, HIVM vessel characteristics depict some evidence of a correlation between tumor response and tumor-associated vessels, which, in the future, this knowledge may be applied to the way we treat tumors

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
