# Peer review of "Advanced Tumor Imaging Approaches in Human Tumors"

_cancers, 2022, doi:10.3390/cancers14061549_

Round 1
Reviewer 1 Report
This review provides a brief introduction of current tumor imaging techniques focusing on artificial intelligence, molecular imaging, and real-time intravital imaging. Authors brought up recent reviews (2018-2021) from a physician’s point of view. Regretfully, the introduction and discussion are full of abstract, monotonous, or obvious statements especially for clinicians engaged in the diagnosis/treatment of cancer. My first impression of this review was its target readers would be students in medical or related school. More in-depth arguments regarding such as strong features, essential difficulties, state of the art and challenges in individual modalities would greatly increase the charm of this manuscript.
Some abbreviations, for example MI for molecular imaging, and MIGS for molecular imaging guided surgery, TME for tumor microenvironment would be a burden for the target readers.
Author Response
This review provides a brief introduction of current tumor imaging techniques focusing on artificial intelligence, molecular imaging, and real-time intravital imaging. Authors brought up recent reviews (2018-2021) from a physician’s point of view. Regretfully, the introduction and discussion are full of abstract, monotonous, or obvious statements especially for clinicians engaged in the diagnosis/treatment of cancer. My first impression of this review was its target readers would be students in medical or related school. More in-depth arguments regarding such as strong features, essential difficulties, state of the art and challenges in individual modalities would greatly increase the charm of this manuscript.
Thank you for this comment. Indeed, our review was meant to provide a simplistic overview of these topics for any level reader. But we have added more content to each section, including the Introduction and Discussion. Additional references have also been added.
Some abbreviations, for example MI for molecular imaging, and MIGS for molecular imaging guided surgery, TME for tumor microenvironment would be a burden for the target readers.
These have been removed as abbreviations.
Reviewer 2 Report
This short review pinpoints the most recent advances in three areas of cancer imaging, namely artificial intelligence, molecular imaging and intravital microscopy, that have the prospect to significantly enhance the efficacy of cancer management in the future in terms of tumor detection, therapy monitoring and overall treatment planning.
The paper is clear and well-written and the reader can easily get a grasp of the latest advancements in tumor imaging approaches. For each approach, the major findings of one source are presented that exemplifies the progress in the field. Overall, the manuscript is suitable for publication although there are a few suggestions/corrections that could be addressed before publishing:
• lines 122-123 ‘‘...made up of ligands (peptides or antibodies)...’’ change to ‘‘made up of target-specific molecules (e.g. small molecules, peptides, antibodies)’’. Currently, most of the clinically used imaging agents are small molecules
• lines 125-127. Please rephrase this sentence since there is a big debate about which imaging modality is more accurate on its own. The current consensus is that combining PET and MRI is indeed advantageous
• There is a huge effort in the field of Molecular Imaging to develop imaging probes that specifically seek the molecular target that is interwoven with the manifestation of disease. This has the prospect to enable the non-invasive molecular characterization of the tumor that can facilitate personalized medicine by devising the optimal treatment planning for the patient. I believe that this should be briefly mentioned in the section of the manuscript that refers to Molecular Imaging.
Author Response
This short review pinpoints the most recent advances in three areas of cancer imaging, namely artificial intelligence, molecular imaging and intravital microscopy, that have the prospect to significantly enhance the efficacy of cancer management in the future in terms of tumor detection, therapy monitoring and overall treatment planning.
The paper is clear and well-written and the reader can easily get a grasp of the latest advancements in tumor imaging approaches. For each approach, the major findings of one source are presented that exemplifies the progress in the field. Overall, the manuscript is suitable for publication although there are a few suggestions/corrections that could be addressed before publishing:
- lines 122-123 ‘‘...made up of ligands (peptides or antibodies)...’’ change to ‘‘made up of target-specific molecules (e.g. small molecules, peptides, antibodies)’’. Currently, most of the clinically used imaging agents are small molecules
We thank the reviewer for these comments. This has been changed as suggested.
- lines 125-127. Please rephrase this sentence since there is a big debate about which imaging modality is more accurate on its own. The current consensus is that combining PET and MRI is indeed advantageous
This has also been changed as suggested.
- There is a huge effort in the field of Molecular Imaging to develop imaging probes that specifically seek the molecular target that is interwoven with the manifestation of disease. This has the prospect to enable the non-invasive molecular characterization of the tumor that can facilitate personalized medicine by devising the optimal treatment planning for the patient. I believe that this should be briefly mentioned in the section of the manuscript that refers to Molecular Imaging.
We agree and have added a new paragraph with several lines of text to this section.
Round 2
Reviewer 1 Report
The manuscript is now more concise, forming a brief introduction of the techniques to clinicians.